

# Topological Frustration can modify
# the nature of a Quantum Phase Transition

**Vanja Marić[1,2], Gianpaolo Torre[3], Fabio Franchini[1] and Salvatore Marco Giampaolo[1]**

**1** Division of Theoretical Physics, Ruđer Bošković Institute,
Bijenička cesta 54, 10000 Zagreb, Croatia
**2** SISSA and INFN, via Bonomea 265, 34136 Trieste, Italy
**3** Department of Physics, Faculty of Science, University of Zagreb,
Bijenička cesta 32, 10000 Zagreb, Croatia

## Abstract

Ginzburg-Landau theory of continuous phase transitions implicitly assumes that microscopic changes are negligible in determining the thermodynamic properties of the system. In this work we provide an example that clearly contrasts with this assumption. We show that topological frustration can change the nature of a second order quantum phase transition separating two different ordered phases. Even more remarkably, frustration is triggered simply by a suitable choice of boundary conditions in a 1D chain. While with every other BC each of two phases is characterized by its own local order parameter, with frustration no local order can survive. We construct string order parameters to distinguish the two phases, but, having proved that topological frustration is capable of altering the nature of a system's phase transition, our results pose a clear challenge to the current understanding of phase transitions in complex quantum systems.



Ginzburg-Landau Theory (GLT) [1–3] is one of the pillars of modern many-body physics. It constitutes an enormous reductionist criterion since it states that phases separated by a phase transition can be distinguished by a *local order parameter*, thus simplifying the macroscopic description of a system with many interacting degrees of freedom into a single emergent property. The characterization of a system's phase through its macroscopic order (or lack thereof) is intimately related to the concept of a phase transition since such a collective property can be changed only through a discontinuity in which the system is reorganized. In principle, one can have a critical point that does not correspond to a rearrangement of the local order, i.e. a transition separating two different disordered phases. However, such occurrences have never been observed. Not in classical systems, but not even in quantum ones. In fact, when the latter challenged standard GLT, it was quickly realized that it can be saved by extending the concept of local order parameter to other types of orders, which are either not strictly local (as, for instance, the nematic order [4,5]), or global (as is the case of topological order [6,7]). Thus, even in the quantum cases, it is expected that a critical point separates two phases with different macroscopic behaviors.

Indeed, the non-local nature of quantum mechanics continues to surprise and to force us to modify our paradigms to include new effects. One such recent realization is the observation that certain boundary conditions can affect some physical behaviors even in the thermodynamic limit [8–10]. GLT stands on the hypothesis that boundary conditions, due to a finite correlation length, cannot influence the bulk behavior of a system. According with this assumption, the prescription is to take the thermodynamic limit before any observable is calculated. However, no truly infinite system exists in nature, and disregarding the total size as a possibly relevant quantity might throw away part of the physics. In a series of recent works, this was proven to be the case for one-dimensional spin chains with Frustrated Boundary Conditions (FBC), that is, periodic boundary conditions with an odd number of sites in the loop [11–13]. In presence of antiferromagnetic interactions, such conditions induce a topological frustration (TF) that generates a conflict between the locally preferred arrangement of staggered magnetization and the incompatibility of such order with the loop geometry with an odd number of sites. Classically, this is the prototypical example of frustration and it is understood that the conflict is resolved by a single defect, a domain wall, that can be placed anywhere in the chain inducing a massive degeneracy, with the number of lowest energy states growing linearly with the chain length [14, 15].

In the quantum case, this degeneracy is generally lifted and the common picture is that the ground state is a superposition of the domain wall states. Although this picture is largely correct, its consequences have not been appreciated until recently. In fact, because the standard staggered order is not sustainable under FBC, the quantum ground state enforces different patterns. In some cases a mesoscopic ferromagnetic magnetization is realized, vanishing algebraically with the system size [11]. In other cases, the system arranges itself in an almost staggered order in which neighboring sites are non perfectly anti-aligned and over the whole chain the magnitude of the magnetization varies continuously [12]. Essentially, in the first situation, the domain walls interfere destructively with one another, while in the second the ground state can be seen as a superposition of two domain wall waves with different momenta and the macroscopic behavior is the result of their interference. Moreover, introducing local defects, other arrangements that do not reduce to standard AFM become possible, also far away from the defects themselves [16]. Hence, FBC act very differently from other boundary conditions (BC). They can affect the bulk, modifying the local order in unexpected ways and even generating a (first-order) phase transition due to a level crossing that only exists with these BC and marks the change between the different types of local order depicted above [12].

In this work, we point out the existence of an even more surprising effect associated with the TF. Namely, we investigate the case in which the orders on both sides of a second order quantum phase transition are "staggered" and thus both incompatible with the FBC. We show that, in some cases, FBC generates a TF that prevents the emergence of any local order, as quantified by observables spreading over a finite support and breaking a Hamiltonian symmetry and hence the system remains locally disordered across the QPT. Since the scaling dimensions of local observables close to the phase transition are usually one of the most important quantities to determine the universality class at criticality [17], we have that, without local order, the quantum phase transition changes its nature in presence of TF with respect to all other physical situations.

We will illustrate this new phenomenon through the example of an exactly solvable model, but this phenomenology goes beyond it. In fact, as shown in a companion, more mathematical, work [18], the killing of any local order by TF is a rather common occurrence in 1D. The exceptions are related to exact ground state degeneracies with states characterized by particular sequences of momenta. Qualitatively, to understand these points, one can start from the usual picture that interprets the effects of FBC as the introduction of a single delocalized particle excitation in system. Typically, this excitation flips every spin as it travels through the system and

in this way destroys any order. However, if the ground state is a superposition of two traveling excitations, they can interfere constructively if their momenta differ by $\pi$ in the thermodynamic limit (at any finite size, an exact $\pi$ momentum is not allowed by the quantization rules imposed by FBC). Thus, if a system with FBC supports at least two degenerate ground state vectors satisfying the momentum condition above, a finite order can ensue, which is locally similar to the usual staggered one, but varies in an incommensurate way along the chain from a finite value to zero. This is the case, for instance, in certain regions of the phase diagram of the XY chain [12]. The analysis in [18] indicates through numerical empirical evidence that systems with purely topological frustration can easily host ground states with momenta close to $\pm\pi/2$ (thus supporting a finite incommensurate order), while spin chains with larger degree of frustration (for instance, with competing next-to-nearest-neighbor interactions) show more complicated patterns which typically result into vanishing local orders.

Although frustration can prevent the establishment of any local order, we expect that the singularity at the QPT indeed signals a rearrangement of the system, although on lengths scaling like the total system size. Exploiting the analytical solvability of the model in the example we consider, we prove the existence and provide the explicit expressions for string order parameters that replace the local ones in distinguishing the two phases. In this way, we provide a path for an extension of GLT, in the sense that the transition indeed separates different types of global orders. Thus, with TF the transition becomes akin to a topological one, although we are not able to provide a definite characterization in this respect. In fact, contrary to traditional topological transition, here the invariant does not seem to be defined in a "mathematical" space (such as momentum space), but rather in real space, where the invariant can be something like the Toulouse criterion, which counts the even/oddness of AFM interactions over the loop.

*A specific example:* To illustrate this peculiar phenomenon we consider the so-called 2-Cluster Ising model, in which a short-range two-body Ising interaction competes with a cluster term acting simultaneously on four contiguous spins [19]:

$$H = \cos\phi \sum_{j=1}^{N} \sigma_j^x \sigma_{j+1}^x + \sin\phi \sum_{j=1}^{N} \sigma_{j-1}^y \sigma_j^z \sigma_{j+1}^z \sigma_{j+2}^y \tag{1}$$

$$= \cos\phi \sum_{j=1}^{N} \sigma_j^x \sigma_{j+1}^x + \sin\phi \sum_{j=1}^{N} O_j O_{j+1} . \tag{2}$$

Here and in the following $\sigma_j^\alpha$ ($\alpha = x, y, z$) stand for Pauli's operators on the $j$-th spin, $O_j = \sigma_{j-1}^y \sigma_j^x \sigma_{j+1}^y$ is the cluster operator [20] that allows to rewrite the cluster interaction term in a form resembling a two body one, $\phi$ is a parameter that allows to tune the relative weight between the two terms and the periodic boundary conditions imply that $\sigma_{j+N}^\alpha = \sigma_j^\alpha$, as well as $O_{j+N} = O_j$.

Usually, this model displays a second-order phase transition between two different ordered phases [19], depending on whether the Ising or the cluster terms dominate. However, by applying FBC (so, setting $N$ to be an odd number), when the interactions favor an antiferromagnetic alignment, TF sets in and the phase diagram modifies accordingly, as we will discuss and is previewed by the phase diagram in Fig. 1

In addition to the aforementioned quantum phase transition, this model holds an important symmetry, which is pivotal in our construction, namely, the Hamiltonian in eq. (1) is invariant under the transformation $\sigma_j^\alpha \leftrightarrow -\sigma_j^\alpha \ \forall \alpha$, which implies that, defining the parity operators as $\Pi^\alpha = \bigotimes_{l=1}^{N} \sigma_l^\alpha$, we have $[H, \Pi^\alpha] = 0 \ \forall \alpha$. Since we are considering the case of odd $N$, different parity operators anti-commute ($\{\Pi^\alpha, \Pi^\beta\} = 0$ for $\alpha \neq \beta$), implying that each eigenstate of the model is, at least, two-fold degenerate. Indeed, if $|\psi\rangle$ is an eigenstate of $H$



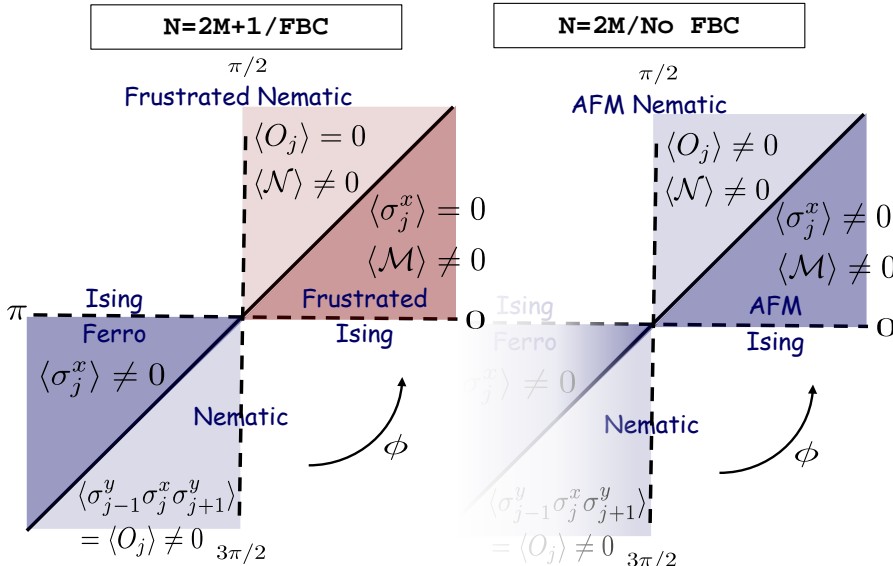

Figure 1: Relevant phase diagram of the 2-cluster-Ising model with Frustrated Boundary Conditions and its comparison with the established phase diagram with other BC.

with $\Pi^z = 1$, the state $\Pi^x |\psi\rangle$ has the same energy, but $\Pi^z = -1$. We stress that this global $SU(2)$ symmetry is quite generic with FBC and is typically violated by the presence of external fields. Its importance is that it allows us to bypass the standard approach for the calculation of order parameters through the application of a symmetry breaking field, since the ground state degeneracy implies that any ground state vector breaks one of the invariances of the Hamiltonian and thus can display a finite magnetization in that direction. We will also use the fact that the mirror symmetry, which is the invariance under the transformation $\sigma_j^\alpha \leftrightarrow \sigma_{2k-j}^\alpha$, where $k$ is the generic site of symmetry, implies that eigenstates either have 0 or $\pi$-momentum, or they appear as degenerate doublets [12].

The Hamiltonian in eq. (1) also enjoys other properties that are convenient, but not necessary, for our analysis. Indeed, this model can be mapped exactly, although non-locally, to a free fermionic systems (see Supplementary Material) and exploiting this fact we can treat larger systems or even get exact analytical results. Moreover, a duality symmetry, consisting in the invariance of the Hamiltonian under the simultaneous exchange $\phi \leftrightarrow \frac{\pi}{2} - \phi$ and $\sigma_j^x \leftrightarrow O_j$, relates the Ising and the nematic phases.

When $\phi \in \left(\frac{3\pi}{4}, \frac{7\pi}{4}\right)$ the dominant interaction favors a ferromagnetic alignment and thus FBC do not induce any frustration, which sets in only in the remaining part of the phase diagram. Here, a double degenerate ground state (due to the global $SU(2)$ symmetry) is separated by a finite energy gap from the other states and in the thermodynamic limit its behavior is indistinguishable from that with open boundary conditions studied in [19]. At $\phi = \frac{5\pi}{4}$ there is a quantum phase transition which separates two differently ordered phases. When the Ising interaction prevails over the cluster one, i.e. for $\phi \in \left(\frac{3\pi}{4}, \frac{5\pi}{4}\right)$, the system shows a ferromagnetic phase characterized by a non-zero value of the magnetization along x. On the other side of the critical point, when $\phi \in \left(\frac{5\pi}{4}, \frac{7\pi}{4}\right)$, we have that the system is in a nematic phase identified by the zeroing of the magnetizations in all directions and the simultaneous rise of a non-vanishing value for the expectation value of the nematic operator $O_j$.

On the contrary, when $\phi \in (0, \frac{\pi}{2})$, both the cluster and Ising interaction are "antiferromagnetic", and hence a TF is induced in the system. Similarly to the phenomenology discussed in [12], the competition between two frustrated interactions increases the ground state degeneracy to four. Denoting by $|p\rangle$ a ground state vector in the odd sector of the $z$-parity with

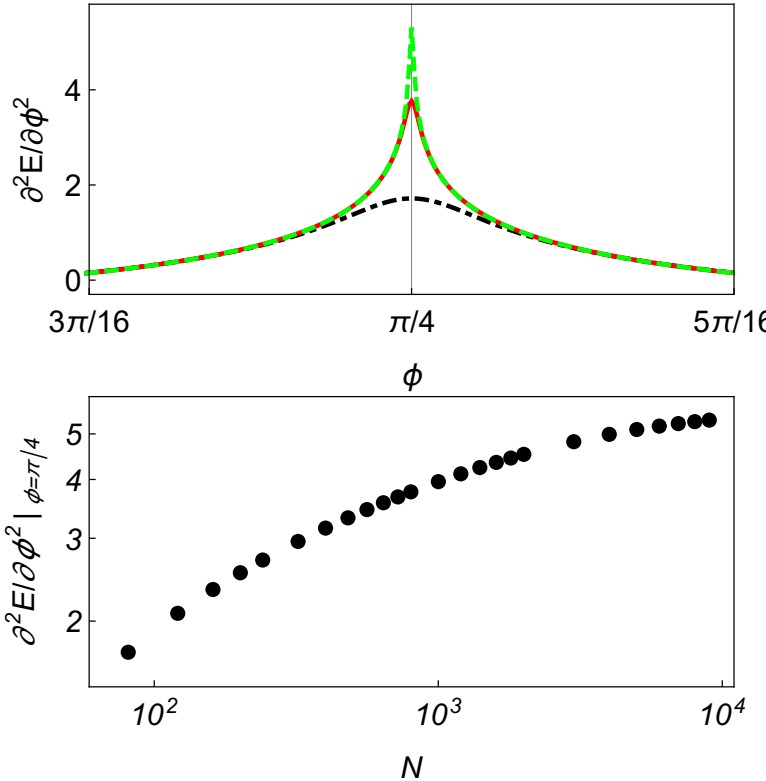

Figure 2: (Color online) - Upper Panel: second derivative of the ground-state energy density (i.e. the energy per site) as function of $\phi$ for different length of the system. $N = 81$ Black dot-dashed line, $N = 801$ Red solid line, $N = 8001$ Green Dashed line. Lower Panel: Dependence of the second derivative of the energy density evaluated for $\phi = \pi/4$ on the size of the system $N$.

lattice momentum $p$, the GS manifold is spanned by four states, two in the odd sector $|\pm p\rangle$, and two in the even one $\Pi^x|\pm p\rangle$. The value of $p$ depends on the value of $N$ mod 8 and takes the value $p \simeq \frac{\pi}{4}$ or $p \simeq \frac{3\pi}{4}$. These states are surmounted by a band (of single particle) states whose gap closes as $1/N^2$ for large $N$.

*Local Order:* As in absence of TF, the ground-state energy displays a critical point when the relative weights of the two interactions coincide, see Fig. 2. On the contrary, several other physical aspects are completely spoiled. To highlight this fact, among all the elements in the manifold, we focus on two of them that capture two different spatial dependences of the order parameters,

$$
\begin{aligned}
|g_1\rangle &= \frac{1}{\sqrt{2}}\big(|p\rangle + \Pi^x|p\rangle\big), \\
|g_2\rangle &= \frac{1}{\sqrt{2}}\big(|p\rangle + \Pi^x|-p\rangle\big).
\end{aligned}
\tag{3}
$$

As we wrote before, in the unfrustrated regimes the role of the order parameters is played by the expectation values of two operators, $\sigma_j^x$ and $O_j$. They share the following properties: 1) they are defined on a finite subset of spins; 2) they commute with $\Pi^x$ and anti-commute with $\Pi^z$. For an operator $K_j$, satisfying both 1) and 2), we have that its expectation value in the state $|g_1\rangle$ reduces to $\langle g_1|K_j|g_1\rangle = \langle p|\Pi^x K_j|p\rangle$ and, due to translational invariance we recover

$$
\langle g_1|K_j|g_1\rangle = \langle p|\Pi^x K_N|p\rangle \quad \forall j.
\tag{4}
$$

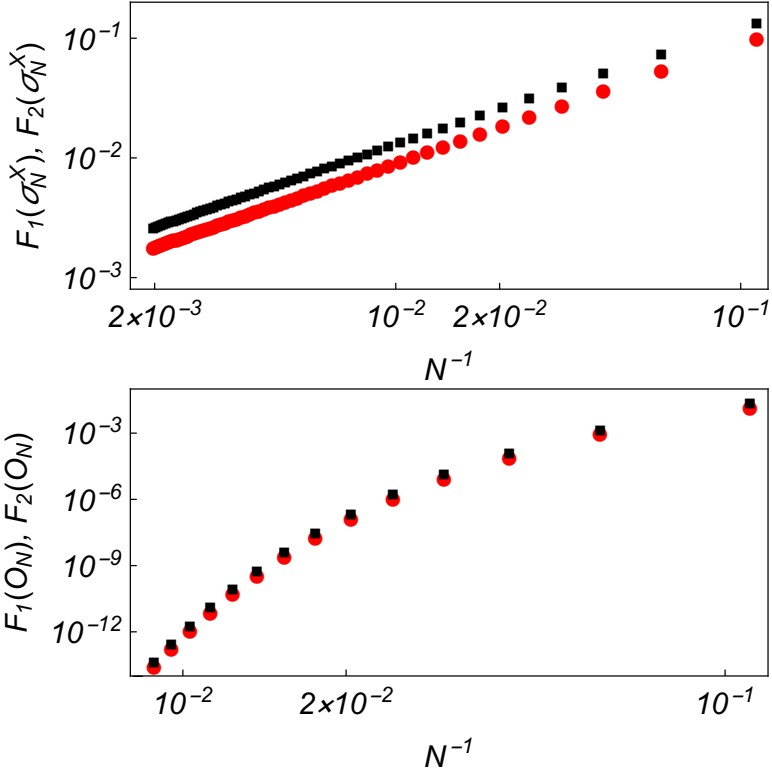

Figure 3: (Color online) - Dependence of $F_1(K_N) = \langle p| \Pi^x K_N |p\rangle$ (red dots) and $F_2(K_N) = \langle -p| \Pi^x K_N |p\rangle$ (black square) on the size of the system $N$ at $\phi = \pi/8$ for different choices of $K_N$: upper panel $K_N = \sigma_N^x$, displaying a power law behavior; lower panel $K_N = O_N = \sigma_{N-1}^y \sigma_N^x \sigma_1^y$, showing an exponential decay. The data runs from $N = 9$ to $N = 505$.

Hence, on $|g_1\rangle$ the order parameters assume the same value on each site of the system. On the contrary, $|g_2\rangle$ is not invariant under spatial translation, and we obtain

$$\langle g_2|K_j|g_2\rangle \ = \ \cos(2jp)\langle -p|\Pi^x K_N |p\rangle \ . \tag{5}$$

Therefore, on $|g_2\rangle$ the order parameters show an incommensurate periodic behavior.

Hence, to study the order parameters in the thermodynamic limit it is enough to analyze the dependence on $N$ of $F_1(K_N) = \langle p|\Pi^x K_N|p\rangle$ and $F_2(K_N) = \langle -p|\Pi^x K_N|p\rangle$. This can be done borrowing the techniques developed in [11, 12] and applied to this case in the supplementary material. The general behavior for the magnetic ($K_N = \sigma_N^x$) and the nematic ($K_N = O_N = \sigma_{N-1}^y \sigma_N^x \sigma_1^y$) order parameter for $\phi \in (0, \frac{\pi}{4})$ is depicted in Fig. 3 as function of the (inverse) size of the system.

In accordance with the unfrustrated case, in the thermodynamic limit we would expect the system to be in a magnetic phase in which either $F_1(\sigma_N^x)$ or $F_2(\sigma_N^x)$ assumes a non-zero value, while both $F_1(O_N)$ and $F_2(O_N)$ vanish. However, while the exponential decay of the nematic order parameter is in agreement with this picture, Fig. 3 clearly shows that, in the thermodynamic limit, there is also no magnetic order since both $F_1(\sigma_N^x)$ and $F_2(\sigma_N^x)$ go to zero linearly with the inverse of the size of the system. Therefore, while in the non-frustrated models the system shows, in the region $\phi \in (0, \frac{\pi}{4})$, an antiferromagnetic order, the introduction of TF in the system induces a zeroing of the magnetic order parameter. In the Supplementary Material we also show that this behavior survives the introduction of an AFM defect in the chain, proving, in accordance with [16], that the phenomenology we discuss is not restricted to purely

translational invariant systems. Moreover, recalling the duality symmetry held by the system, the behavior of the magnetic order parameter for $\phi \in (0, \frac{\pi}{4})$ is mirrored by the nematic order parameter for $\phi \in (\frac{\pi}{4}, \frac{\pi}{2})$. Hence, when FBC are imposed, the order parameters characterizing the two macroscopic phases of the unfrustrated models vanish at both sides of the critical point, making them unable to characterize the phase transition.

But we can go further. Indeed, we can prove that not only the magnetization and the nematic order parameter both vanish in both phases, but that this result extends to any possible local order parameter, i.e. to any expectation value of a local observable that anticommutes with at least one of the parity operators $\Pi^\alpha$. In fact, in [18], we have shown that a wide class of topologically frustrated models, to which the 2-cluster-Ising also belongs, cannot exhibit a finite local order parameter in the vicinity of the classical antiferromagnetic point ($\phi = 0$), unless the difference between the momenta of two ground states tends to $\pm\pi$ in the thermodynamic limit. Since in our case we have that this difference tends to $\pi/2$, the expectation values of all local observables that can play the role of order parameter vanish in the thermodynamic limit close to the point $\phi = 0$ and, hence, we expect that they stay equal to zero until the quantum critical point at $\phi = \pi/4$ is reached. Moreover, applying the duality arguments, it follows that the expectation values of local observables must vanish also in the vicinity of the point $\phi = \pi/2$ and therefore also in the whole region $\phi = (\pi/4, \pi/2)$. As a consequence, since the expectation value of all such local observables vanishes at both sides of the critical point, there is no local order parameter that can characterize the quantum phase transition at either sides. To our knowledge, this is the first case in which topological frustration, and hence a change in the boundary conditions of a system, affects the thermodynamic phase of a spin system so deeply up to completely remove the presence of local order parameters.

*String Order:* However, the result in Ref, [18] does not apply to operators whose support scales with the length of the chain and this fact discloses the possibility that the two macroscopic phases can be distinguished by string order parameters whose presence is normally associated with some kind of topological ordered phases [19–22]. We have not yet been able to identify a strong, geometric criterion to define the string order connected with TF, but, exploiting the microscopical structure of the model under consideration, we have indeed succeeded in constructing two string operators that suit our needs, namely:

$$\mathcal{M} = \prod_{k=1}^{I(N)} \left( \sigma_{4k-2}^x \sigma_{4k-1}^x \right); \quad \mathcal{N} = \prod_{k=1}^{I(N)} \left( O_{4k-2} O_{4k-1} \right), \tag{6}$$

where $I(N)$ depends on the length of the chain and it is equal to $\frac{N-1}{4}$ for $N \bmod 4 = 1$ and to $\frac{N+1}{4} - 1$ in case of $N \bmod 4 = 3$. Both operators commute with all the parity operators $\Pi^\alpha$. It is easy to see that, defining $\mathcal{F}_{1,2}(\mathcal{K}) = \langle g_{1,2} | \mathcal{K} | g_{1,2} \rangle$ for a generic string operator $\mathcal{K}$ for which $[\mathcal{K}, \Pi^\alpha] = 0 \ \forall \alpha$, we have $\mathcal{F}_1(\mathcal{K}) = \mathcal{F}_2(\mathcal{K})$.

These expectation values can be studied analytically using the asymptotic properties of determinants studied in [13] (see also the Supplementary Material), and for $\mathcal{K} = \mathcal{M}, \mathcal{N}$ in the region $\phi \in (0, \pi/4)$ we obtain

$$\mathcal{F}_1(\mathcal{M}) \overset{N \to \infty}{\simeq} (-1)^{I(N)} \frac{1}{2} (1 - \tan^2 \phi)^{\frac{1}{4}},$$

$$\mathcal{F}_1(\mathcal{N}) \overset{N \to \infty}{\simeq} 0. \tag{7}$$

In the same region the finite-size results for $\mathcal{F}_1(\mathcal{M})$ and $\mathcal{F}_1(\mathcal{N})$ are depicted in Fig. 4, where it can be seen that the second goes to zero algebraically with the system size. In the thermodynamic limit the expectation value of the string operator goes continuously to zero at $\phi = \pi/4$, and can thus serve to characterize the quantum phase transition. This picture of the continuous quantum phase transition is coherent with the one inferred from the second

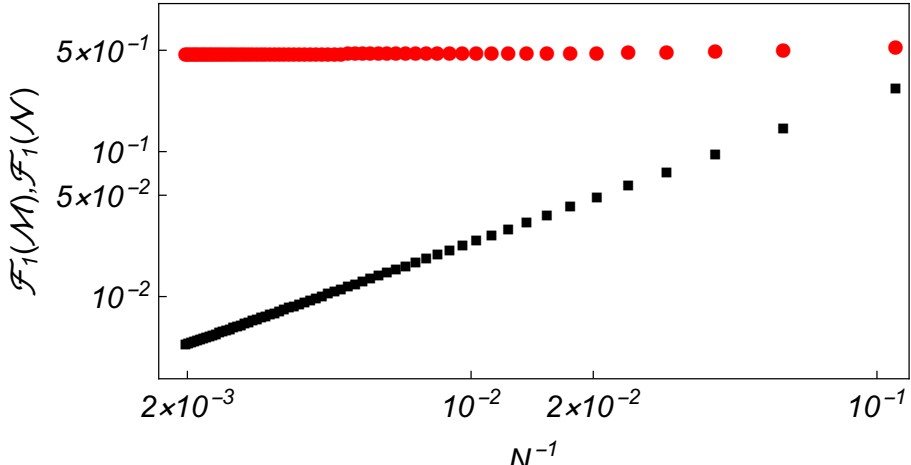

Figure 4: (Color online) - Dependence of $\mathcal{F}_1(\mathcal{M}) = \langle g_1 | \mathcal{M} | g_1 \rangle$ (Red dots) and $\mathcal{F}_1(\mathcal{N}) = \langle g_1 | \mathcal{N} | g_1 \rangle$ (Black squares) on the size of the system $N$ at $\phi = \pi/8$. The data runs from $N = 9$ to $N = 505$ and shows the power law decay of one of the string order parameters, while the other remains finite in the thermodynamic limit.

derivative of the ground state energy. Of course, taking into account that $\mathcal{M}$ and $\mathcal{N}$ are one image of the other under the duality transformation, their behavior is mirrored in the region $\phi \in (\pi/4, \pi/2)$.

*Conclusions:* We have shown how the presence of a TF, induced by the application of FBC in a system with antiferromagnetic interactions, can change the nature of the phase transition in a one-dimensional spin system. While in absence of such kind of frustration, the phases at different sides of the critical point admit two different (staggered) local order parameters, when TF comes into play such quantities vanish in the thermodynamic limit. And this fact is not limited to the expectation value of the operators associated to the order parameters without frustration, but extends to all the operators that can act as local order parameters. We have focused on the example of the 2-Cluster-Ising chain to illustrate this phenomenology, but the vanishing of local order with FBC is common in systems with interactions beyond nearest neighbor which induce additional frustration [18].

These results lead to a deep rethinking within the standard approach to phase transitions. Indeed, usually, approaching a phase transition, only one length scale becomes important: that of the (dominant) correlation length (or the inverse mass gap). This is because the system size is already considered bigger than any other length scale and thus irrelevant. However, this is an idealization, since in practice it is much easier to reach a very large correlation length than to have a truly infinite system. Thus, approaching a critical point, it would be important to also consider scaling quantities that include the size of the system. While under all other boundary conditions this turns out not to be necessary, in presence of FBC this is not the case, as also already pointed out in Ref. [23], and the system size suppresses any local order. Nonetheless, the divergences of the correlation length that causes the discontinuity in the second derivative of the free energy can still be harvested to define an order (and disorder) parameter with different behaviors across the transition, but requires an observable spreading through the whole loop, that is, a string order. While we cannot guarantee that the string parameters that we defined in eq. (6) are the optimal ones to classify the phases, nor we can provide an interpretation for what they represent (indeed more work is required in this respect) in this way, we have shown that the traditional elements of GLT mix differently in the presence of TF and provide a completely unexpected phenomenology.

Before concluding, we wish to underline that our results are, to a large extent, resilient

to the presence of a localized defect in the Hamiltonian, as we show in the Supplementary Material, thus proving that the phenomenon we discussed in the present paper cannot be considered simply as resulting from fine-tuning in the system parameters and thus our prediction can be tested under experimental conditions.

## Acknowledgments

We acknowledge support from the European Regional Development Fund – the Competitiveness and Cohesion Operational Programme (KK.01.1.1.06 – RBI TWIN SIN) and from the Croatian Science Foundation (HrZZ) Projects No. IP–2016–6–3347 and IP–2019–4–3321. SMG, FF, and GT also acknowledge support from the QuantiXLie Center of Excellence, a project co–financed by the Croatian Government and European Union through the European Regional Development Fund – the Competitiveness and Cohesion (Grant KK.01.1.1.01.0004).

## Supplementary Material

### Diagonalization of the Hamiltonian

Here we diagonalize the Hamiltonian of the 2-Cluster Ising Model eq. (1)

$$
\begin{aligned}
H &= \cos\phi \sum_{j=1}^{N} \sigma_j^x \sigma_{j+1}^x + \sin\phi \sum_{j=1}^{N} \sigma_{j-1}^y \sigma_j^z \sigma_{j+1}^z \sigma_{j+2}^y \\
&= \cos\phi \sum_{j=1}^{N} \sigma_j^x \sigma_{j+1}^x + \sin\phi \sum_{j=1}^{N} O_j O_{j+1} ,
\end{aligned}
$$

with a particular emphasis in the parameter range $\phi \in (0, \pi/2)$. The Hamiltonian can be diagonalized exploiting the Jordan–Wigner transformation [24, 25]

$$
c_j = \left( \bigotimes_{l=1}^{j-1} \sigma_l^z \right) \frac{\sigma_j^x + \iota\sigma_j^y}{2}, \quad c_j^\dagger = \left( \bigotimes_{l=1}^{j-1} \sigma_l^z \right) \frac{\sigma_j^x - \iota\sigma_j^y}{2}, \tag{8}
$$

that maps spins into spinless fermions. We split the Hilbert space and the Hamiltonian in the two $\Pi^z$ sectors as

$$
H = \frac{1+\Pi^z}{2} H^+ \frac{1+\Pi^z}{2} + \frac{1-\Pi^z}{2} H^- \frac{1-\Pi^z}{2}, \tag{9}
$$

and in each sector we Fourier transform the fermionic operators

$$
b_q = \frac{1}{\sqrt{N}} \sum_{j=1}^{N} c_j \, e^{-\iota q j}, \quad b_q^\dagger = \frac{1}{\sqrt{N}} \sum_{j=1}^{N} c_j^\dagger \, e^{\iota q j}, \tag{10}
$$

using two sets of momenta $q$, depending on the parity,. We take $q \in \Gamma^+ = \{\frac{2\pi}{N}(k + \frac{1}{2})\}$ for the even parity sector ($\Pi^z = 1$) and $q \in \Gamma^- = \{\frac{2\pi}{N}k\}$ for the odd one ($\Pi^z = -1$), with $k$ running over all integers between 0 and $N - 1$ in both cases. Then we make the Bogoliubov rotation

$$
a_q = \cos\theta_q \, b_q + \iota \sin\theta_q \, b_{-q}^\dagger, \tag{11}
$$

with the Bogoliubov angle defined as

$$
\theta_q = \tan^{-1} \frac{|\sin\phi + \cos\phi \, e^{\iota 4q}| - \sin\phi \cos(3q) - \cos\phi \cos q}{-\sin\phi \sin(3q) + \cos\phi \sin q} \tag{12}
$$

for $q \neq 0, \pi$ and by $\theta_0 = \theta_\pi = 0$. The Bogoliubov angle also satisfies

$$e^{\iota 2\theta_q} = e^{\iota q} \frac{\cos\phi + \sin\phi \ e^{-\iota 4q}}{|\cos\phi + \sin\phi \ e^{-\iota 4q}|}. \tag{13}$$

Through these series of exact, non-local transformations, the Hamiltonian in each sector is brought to the free-fermionic form

$$H^{\pm} = \sum_{q \in \Gamma^{\pm}} \varepsilon_q \left( a_q^{\dagger} a_q - \frac{1}{2} \right), \tag{14}$$

in terms of the the Bogoliubov operators, where the energies $\varepsilon_q$ associated to each mode with momentum $q \in \Gamma^{\pm}$ are given by

$$\begin{aligned}
\varepsilon_q &= 2|\cos\phi + \sin\phi \ e^{\iota 4q}| &&\forall q \neq 0, \pi; \\
\varepsilon_0 &= 2(\cos\phi + \sin\phi) && q = 0 \in \Gamma^-; \\
\varepsilon_\pi &= -2(\cos\phi + \sin\phi) && q = \pi \in \Gamma^+.
\end{aligned} \tag{15}$$

Depending on the value of $\phi$, the energies of the modes with $q = 0 \in \Gamma^-$ and $q = \pi \in \Gamma^+$ are different from the others because they can become negative. Without frustration, these negative energy modes are responsible for the ground state degeneracy that allows for the spontaneous symmetry breaking mechanism, while in presence of frustration they play a different and pivotal role in the emerging phenomenolgy. Indeed, for each $\phi$ the ground states of the system can be determined starting from the vacuum of Bogoliubov fermions in the two sectors ($|0^{\pm}\rangle$), which, by construction, have positive parity $\Pi^z = 1$, and taking into account both the presence of modes with negative energy and the parity constrains.

When $\phi \in (-\pi, -\frac{\pi}{2})$ (both interactions are "ferromagnetic"), we have $\varepsilon_\pi > 0$ while $\varepsilon_0 < 0$ and hence, in each parity sector, the state with the lowest energy, respectively $|0^+\rangle$ and $a_0^{\dagger}|0^-\rangle$, fulfills the parity requirement. They are separated from the other states by a finite energy gap that, in the thermodynamic limit becomes equal to $-2\varepsilon_0 = 2\varepsilon_\pi \neq 0$. This is the same physical picture that can be found also assuming open boundary conditions [19] and hence also the thermodynamic behavior is the same. A quantum phase transition at $\phi = -3\frac{\pi}{4}$ separates two different ordered phases. When the Ising interaction prevails over the cluster one, i.e. for $\phi \in (-\pi, -3\frac{\pi}{4})$, the system shows a ferromagnetic phase characterized by a non-zero value of the magnetization along x. On the other side of the critical point, when $\phi \in (-3\frac{\pi}{4}, -\frac{\pi}{2})$, we have that the system is in a nematic phase identified by the zeroing of the magnetization in all directions and the simultaneous setting up of a non-vanishing value of the expectation value of the nematic operator $O_j$.

On the contrary, when $\phi \in (0, \frac{\pi}{2})$, both the cluster and Ising interaction are "antiferromagnetic", and hence we have TF in the system: in this region $\varepsilon_0 > 0$ while $\varepsilon_\pi < 0$. As a consequence of this, the two states with the lowest energy are, respectively, $a_\pi^{\dagger}|0^+\rangle$ in the even sector and $|0^-\rangle$ in the odd one. Both of them violate the parity requirements in (9) and, therefore, cannot be eigenstates of the the Hamiltonian in eq. (1). Instead, the ground-states belong to a four-fold degenerate manifold spanned by the states $|\pm p\rangle \equiv a_{\pm p}^{\dagger}|0^-\rangle$ in the odd sector and $\Pi^x|\pm p\rangle$ in the even one, where the momentum $p$ obeys

$$p = \begin{cases}
\frac{\pi}{4} - \frac{\pi}{4N}, & \text{N mod 8=1} \\
\frac{3\pi}{4} - \frac{\pi}{4N}, & \text{N mod 8=3} \\
\frac{3\pi}{4} + \frac{\pi}{4N}, & \text{N mod 8=5} \\
\frac{\pi}{4} + \frac{\pi}{4N}, & \text{N mod 8=7}.
\end{cases} \tag{16}$$

These states are surmounted by a band states whose gap closes as $1/N^2$ for large $N$.

**The order parameters and their representation as determinants**

In evaluating the order parameters it is useful to first define the Majorana fermions

$$A_j = \Big(\bigotimes_{l=1}^{j-1} \sigma_l^z\Big)\sigma_j^x, \quad B_j = \Big(\bigotimes_{l=1}^{j-1} \sigma_l^z\Big)\sigma_j^y. \tag{17}$$

The expectation values of interest will be expressed in terms of determinants of Majorana correlation matrices, employing techniques similar to the ones used in [12] for the magnetization. Among the states in the ground space manifold a special role is played by two of them, defined as

$$|g_\pm\rangle \equiv \frac{1}{\sqrt{2}}(|p\rangle \pm |-p\rangle), \tag{18}$$

which are eigenstates of $\Pi^z$ with eigenvalue $-1$ and of the mirror operator with respect to site $N$, denoted by $M_N$, [12] with eigenvalue $\pm 1$ (respectively). Note that the site $N$ is defined with respect to the beginning of the Jordan-Wigner string in eq. (8).

As discussed in the main text, for an operator $K_N$, that commutes with $\Pi^x$ and anticommutes with $\Pi^z$, the ground state expectation values depend on the matrix elements $F_1(K_N) = \langle p|\Pi^x K_N|p\rangle$ and $F_2(K_N) = \langle -p|\Pi^x K_N|p\rangle$. Assuming $M_N K_N M_N = K_N$, we have both $\langle p|K_N \Pi^x|p\rangle = \langle -p|K_N \Pi^x|-p\rangle$ and $\langle -p|K_N \Pi^x|p\rangle = \langle p|K_N \Pi^x|-p\rangle$, so the matrix elements can be expressed through the expectation values of the states in eq. (18) as

$$\begin{aligned} F_1(K_N) &= \tfrac{1}{2}(\langle g_+|K_N\Pi^x|g_+\rangle + \langle g_-|K_N\Pi^x|g_-\rangle), \\ F_2(K_N) &= \tfrac{1}{2}(\langle g_+|K_N\Pi^x|g_+\rangle - \langle g_-|K_N\Pi^x|g_-\rangle). \end{aligned} \tag{19}$$

Now, because the operator $K_N\Pi^x$ commutes with $\Pi^z$, the expectation values $\langle g_\pm|K_N\Pi^x|g_\pm\rangle$ can be obtained following a well known approach that applies to all operators that commute with $\Pi^z$ and all ground states of well-defined parity [25, 26].

The first step is to express $K_N\Pi^x$ as a product of Majorana fermions. The second step is to use Wick theorem (the same argument as in [12] can be used to justify the validity of Wick theorem in these states) to express the expectation values as determinants of matrices of two-point Majorana correlators. Adopting the short notation $\langle\cdot\rangle_\pm = \langle g_\pm|\cdot|g_\pm\rangle$, we have that $\langle A_j A_k\rangle_\pm = \langle B_j B_k\rangle_\pm = \delta_{jk}$ and

$$\begin{aligned} -\imath\langle A_j B_k\rangle_\pm &= \frac{1}{N}\sum_{q\in\Gamma^-}e^{i2\theta_q}e^{-iq(j-k)} \\ &\quad -\frac{2}{N}\cos\big[p(j-k)-2\theta_p\big]\mp\frac{2}{N}\cos\big[(j+k)p\big]. \end{aligned} \tag{20}$$

**Local Order Parameter:** Following this approach, for $K_N = \sigma_N^x$ we get

$$\langle\sigma_N^x\Pi^x\rangle_\pm = (-1)^{\frac{N-1}{2}}\det\mathbf{C}^{(1)}, \tag{21}$$

where the elements of the $\frac{N-1}{2}\times\frac{N-1}{2}$ matrix $\mathbf{C}^{(1)}$ are equal to $\mathbf{C}^{(1)}_{\alpha,\beta} = -\imath\langle A_{2\alpha}B_{2\beta-1}\rangle_\pm$, for $\alpha,\beta\in\{1,2,\dots(N-1)/2\}$. Similarly, for $K_N = O_N$ we obtain

$$\langle O_N\Pi^x\rangle_\pm = (-1)^{\frac{N-1}{2}}\det\mathbf{C}^{(2)}, \tag{22}$$

where $\mathbf{C}^{(2)}$ is an $\frac{N+1}{2}\times\frac{N+1}{2}$ matrix. Its elements are given by $\mathbf{C}^{(2)}_{\alpha,\beta} = -\imath\langle A_{f(\alpha)}B_{f'(\beta)}\rangle_\pm$, where as $\alpha,\beta$ go over $1,2,\dots(N+1)/2$ we have that $f(\alpha)$ and $f'(\beta)$ assume the values $f(\alpha) = 1,3,5,\dots,N-4,N-2,N-1$ and $f'(\beta) = 1,2,4,6,\dots,N-3,N-1$.

**String Order Parameters** To define string operators that can be exploited to characterize this quantum phase transition we started from an observation made in Ref. [21]. In that work, the authors prove that in the thermodynamic limit of the unfrustrated cluster-Ising model, the only Majorana correlation functions that are not zero are those for which the site indices satisfy the relation $i - j = 3k - 1$ where $k$ is an integer. Even if this observation was made for a different model and in the absence of frustration, from the expression of the Majorana functions it is easy to observe that a similar property holds also in our case. In fact, it is possible to see that, in our case, all the Majorana correlation functions that do not satisfy the property $i - j = 4k - 1$ vanish in the limit of large $N$. The presence of TF adds $1/N$ corrections to Majorana fermions so it does not affect these properties, but it can affect the values of order parameters because they are expressed in terms of determinants of Majorana correlation matrices whose size grows with $N$. Hence, we tried to see if string operators, whose expectation value depends only on Majorana correlators that do not become zero in the thermodynamic limit, are able to characterize the quantum phase transition. Among all the possibilities we focus on two of them that are one the image of the other after the duality transformation, namely

$$\mathcal{M} = \prod_{k=1}^{I(N)} \left( \sigma_{4k-2}^x \sigma_{4k-1}^x \right); \quad \mathcal{N} = \prod_{k=1}^{I(N)} \left( O_{4k-2} O_{4k-1} \right), \tag{23}$$

where $I(N) = \frac{N-1}{4}$ for $N \bmod 4 = 1$ and $I(N) = \frac{N+1}{4} - 1$ for $N \bmod 4 = 3$.

We can use the same approach as for the local operators for the evaluation of the expectation values for two string order parameters, $\mathcal{F}_1(\mathcal{M})$ and $\mathcal{F}_1(\mathcal{N})$, in terms of determinants. With respect to the states $|g_\pm\rangle$ they read

$$\mathcal{F}_1(\mathcal{K}) = \frac{1}{2} \left( \langle \mathcal{K} \rangle_+ + \langle \mathcal{K} \rangle_- \right), \tag{24}$$

with $\mathcal{K} = \mathcal{N}, \mathcal{M}$. In this case we have the determinant representation

$$\begin{aligned} \langle \mathcal{M} \rangle_\pm &= (-1)^{I(N)} \det \mathbf{C^{(3)}}, \\ \langle \mathcal{N} \rangle_\pm &= (-1)^{I(N)} \det \mathbf{C^{(4)}}, \end{aligned} \tag{25}$$

where $\mathbf{C^{(3)}}$ and $\mathbf{C^{(4)}}$ are $I(N) \times I(N)$ matrices, with $I(N) = \frac{N-1}{4}$ for $N \bmod 4 = 1$ and $I(N) = \frac{N+1}{4} - 1$ for $N \bmod 4 = 3$. Their elements are given by $\mathbf{C^{(3)}}_{\alpha,\beta} = -\iota \langle A_{4\alpha-1} B_{4\beta-2} \rangle_\pm$ and $\mathbf{C^{(4)}}_{\alpha,\beta} = -\iota \langle A_{4\alpha} B_{4\beta-3} \rangle_\pm$, for $\alpha, \beta \in \{1, 2, \dots, I(N)\}$.

**Analytic evaluation of the string order parameter:** The expressions in eq. (24) are efficient for the numerical evaluation of the string order parameters. Here we show how to analytically evaluate the value that the string order parameter $\mathcal{F}_1(\mathcal{M})$ assumes in the thermodynamic limit. To do so we express it in terms of Toeplitz determinants, using an approach similar to the one used in [13,27] for other quantities. The string order parameter $\mathcal{F}_1(\mathcal{N})$ can be studied in an analogous way.

We start by noting that the string order parameter is equal to

$$\mathcal{F}_1(\mathcal{M}) = (-1)^{I(N)} \langle 0^- | a_p \prod_{k=1}^{I(N)} (-\iota A_{4k-1} B_{4k-2}) a_p^\dagger | 0^- \rangle \tag{26}$$

and then we make Wick contractions in the vacuum state $|0^-\rangle$. Adopting the short notation $\langle \cdot \rangle_0 = \langle 0^- | \cdot | 0^- \rangle$, we have $\langle A_j A_k \rangle_0 = \langle B_j B_k \rangle_0 = \delta_{jk}$ and

$$-\iota \langle A_j B_k \rangle_0 = \frac{1}{N} \sum_{q \in \Gamma^-} e^{i2\theta_q} e^{-iq(j-k)}. \tag{27}$$

Moreover, since we can express the Majorana fermions as

$$A_j = \frac{1}{\sqrt{N}} \sum_{q \in \Gamma^-} (a_q^\dagger + a_{-q}) e^{\iota \theta_q} e^{-\iota q j},$$
$$-\iota B_j = \frac{1}{\sqrt{N}} \sum_{q \in \Gamma^-} (a_q^\dagger - a_{-q}) e^{-\iota \theta_q} e^{-\iota q j}, \tag{28}$$

we can easily find the contractions

$$-\iota \langle a_p A_j \rangle_0 \langle B_k a_p^\dagger \rangle_0 = -\frac{1}{N} e^{\iota 2\theta_p} e^{-\iota p(j-k)},$$
$$-\iota \langle a_p B_k \rangle_0 \langle A_j a_p^\dagger \rangle_0 = \frac{1}{N} e^{-\iota 2\theta_p} e^{\iota p(j-k)}. \tag{29}$$

Performing all the Wick contractions in eq. (26) and using the basic properties of determinants, the string order parameter can be expressed as

$$\mathcal{F}_1(\mathcal{M}) = (-1)^{I(N)} \big[ \big( \det \tilde{\mathbf{C}} + \text{c.c.} \big) - \det \mathbf{C} \big], \tag{30}$$

where $\tilde{\mathbf{C}}$ and $\mathbf{C}$ are $I(N) \times I(N)$ matrices with the elements

$$\mathbf{C}_{\alpha,\beta} = -\iota \langle A_{4\alpha-1} B_{4\beta-2} \rangle_0,$$
$$\tilde{\mathbf{C}}_{\alpha,\beta} = \mathbf{C}_{\alpha,\beta} - \frac{1}{N} e^{\iota(2\theta_p - p)} e^{-\iota 4p(\alpha-\beta)}, \tag{31}$$

for $\alpha, \beta \in \{1, 2, \dots, I(N)\}$, which give an alternative expression to eq. (24) for its analytical evaluation.

Using eq. (13), approximating the sum in eq. (27) by an integral, and doing some simple manipulations we get

$$\mathbf{C}_{\alpha,\beta} \overset{N \to \infty}{\simeq} \int_0^{2\pi} f(e^{\iota\theta}) e^{-\iota\theta(\alpha-\beta)} \frac{d\theta}{2\pi}, \tag{32}$$

where

$$f(e^{\iota\theta}) = \frac{1 + \tan\phi \; e^{-\iota\theta}}{|1 + \tan\phi \; e^{-\iota\theta}|}. \tag{33}$$

With this definition we can also write

$$\tilde{\mathbf{C}}_{\alpha,\beta} = \mathbf{C}_{\alpha,\beta} - \frac{1}{N} f(e^{\iota\theta_0}) e^{-\iota\theta_0(\alpha-\beta)}, \tag{34}$$

where $\theta_0 = 4p$.

Thus for $\phi \in (0, \pi/4)$ the matrix $\mathbf{C}$ is a standard Toeplitz matrix, whose symbol is a non-zero analytic function in an annulus around the unit circle, with zero winding number. Its determinant can be computed in the standard way using strong Szegő limit theorem (see [28]) and we get

$$\det \mathbf{C} \overset{N \to \infty}{\simeq} (1 - \tan^2 \phi)^{\frac{1}{4}}. \tag{35}$$

To compute the determinant of $\tilde{\mathbf{C}}$ we need to use Theorem 1 from [13], which gives a correction to Szegő theorem for this type of Toeplitz matrices. We get

$$\det \tilde{\mathbf{C}} \overset{N \to \infty}{\simeq} \left(1 - \frac{I(N)}{N}\right) \det \mathbf{C} \overset{N \to \infty}{\simeq} \frac{3}{4}(1 - \tan^2 \phi)^{\frac{1}{4}}. \tag{36}$$

Finally, from eq. (30) we get the string order parameter

$$\mathcal{F}_1(\mathcal{M}) \overset{N \to \infty}{\simeq} (-1)^{I(N)} \frac{1}{2}(1 - \tan^2 \phi)^{\frac{1}{4}}. \tag{37}$$

For $\phi \in (\pi/4, \pi/2)$ the symbol $f$ has a non-zero winding number. It follows immediately from Theorem 2 in [13] that the string order parameter is zero in the thermodynamic limit,

$$\mathcal{F}_1(\mathcal{M}) \overset{N \to \infty}{\simeq} 0. \tag{38}$$

**Effects of the presence of a defect**

The scenario that we have depicted for the 2-cluster-Ising model is very peculiar, and it is normal to wonder whether it is resilient to the presence of noise, or it is the result of fine-tuning in the system parameters. Obviously, a complete analysis of the effects of the presence of defects in our model is far beyond the scope of this paper and has been the subject of analysis in [16], but for a different model. Here we discuss a simple example that shows that the phenomenology that we have depicted in the main body of this work is quite resilient.

Hence let us take into account the Hamiltonian

$$
\begin{aligned}
H' &= +\sin\phi \sum_{j=1}^{N} \sigma^y_{j-1}\sigma^z_j\sigma^z_{j+1}\sigma^y_{j+2} + \\
&\quad +\cos\phi \sum_{j=1}^{N-1} \sigma^x_j\sigma^x_{j+1} + \cos(\phi+\delta_x)\,\sigma^x_N\sigma^x_1,
\end{aligned}
\tag{39}
$$

that coincides with the Hamiltonian in eq. (1) except for the presence of a defect in the Ising interaction, localized between the first and the last spin of the model. Such a presence implies that the new Hamiltonian in eq. (39) is neither translationally invariant nor preserves the mirror symmetries, except the one with respect to the $(N+1)/2$-th spin, while it continues to commute with all the parity operators. As a consequence, the ground state degeneracy of $H'$ is reduced to two, even in the region where $H$, without the defect, presents a four dimensional manifold. However, independently of the parameters, $H'$ always includes states of both parities so we can continue to use the already described approach to evaluate directly the order parameter.

Since $H'$ is no more translationally invariant, it is now impossible to find an exact analytical expression for the ground states. We are then forced to resort to an efficient numerical procedure based on the fact that the Hamiltonian is, in each $\Pi^z$ sector, still quadratic in terms of the fermionic operators and hence its eigenstates can be found following Ref. [16, 25]. We focus on the odd sector ($\Pi^z = -1$), since having the ground state $|g'_-\rangle$ of $H'$ belonging to the odd sector we can construct the ground state of $H'$ belonging to the even sector as $|g'_+\rangle = \Pi^x |g'_-\rangle$. We write $H'$ in the odd sector as

$$
H' = \sum_{j,k=1}^{N} \left[ c^\dagger_j S_{j,k} c_k + \frac{1}{2}\left(c^\dagger_j T_{j,k} c^\dagger_k + \text{h.c.}\right) \right],
\tag{40}
$$

where the matrices $\mathbf{S} = \mathbf{S}^\dagger$ and $\mathbf{T}^\dagger = -\mathbf{T}$ can be easily obtained by inspection from eq. (39). In this approach, the ground state $|g'_-\rangle$ can be expressed in terms of the vectors $\Phi_k$ and $\Psi_k$, that are the solution of the problem:

$$
\Phi_k(\mathbf{S}-\mathbf{T})(\mathbf{S}+\mathbf{T}) = \Lambda_k^2 \Phi_k,
\tag{41}
$$

$$
\Phi_k(\mathbf{S}-\mathbf{T}) = \Lambda_k \Psi_k,
\tag{42}
$$

with the eigenvalues $\Lambda_k^2$ sorted in descending order. From the knowledge of $\Phi_k$ and $\Psi_k$ it is easy to recover the correlation functions of the Majorana operators. With respect to the odd-sector ground state we have $\langle g'_-|A_jA_k|g'_-\rangle = \langle g'_-|B_jB_k|g'_-\rangle = \delta_{jk}$ and

$$
-\imath\langle g'_-|A_jB_k|g'_-\rangle = \sum_{l=1}^{N-1}\Psi_{lj}\Phi_{lk}.
\tag{43}
$$

If we consider the ground state choice

$$
|g'\rangle = \frac{1}{\sqrt{2}}(|g'_-\rangle + |g'_+\rangle) = \frac{1}{\sqrt{2}}(|g'_-\rangle + \Pi^x|g'_-\rangle),
\tag{44}
$$

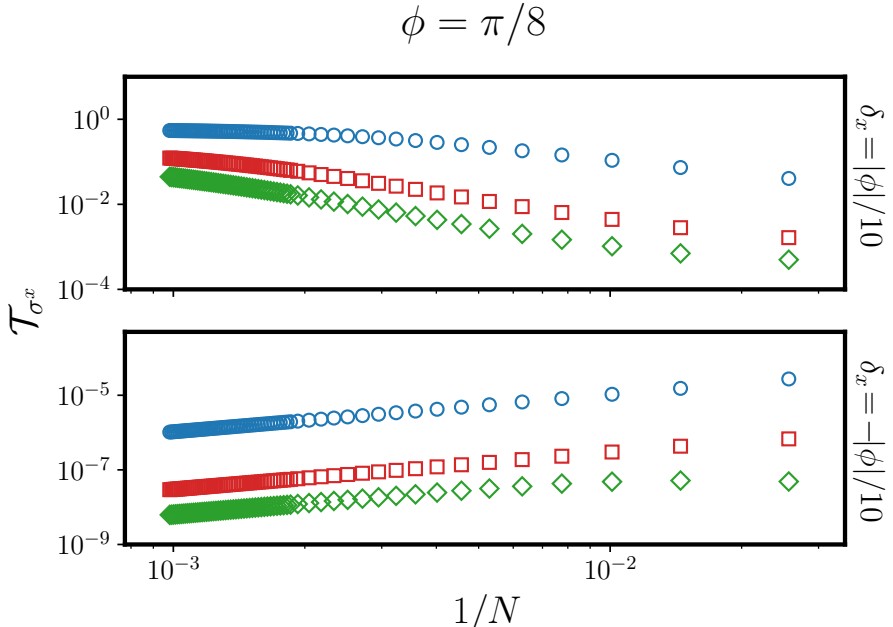

Figure 5: (Color online) Absolute value of the Discrete Fourier transform (DFT) of the magnetization $\langle g'|\sigma_j^x|g'\rangle$ at $\phi = \frac{\pi}{8}$, as a function of the inverse chain length, for chain lengths up to $N = 1019$. Data corresponds to the following different momenta: green diamonds $k = \frac{N\pm5}{2}$, red squares $k = \frac{N\pm3}{2}$, and blue circles $k = \frac{N\pm1}{2}$. A ferromagnetic type defect (upper panel) yields a staggered AFM order, while the presence of an antiferromagnetic one (lower panel) gives rise to an algebraic decay of the magnetization, characteristic to the presence of TF (see the text for discussion).

we obtain that the site dependent expectation values of the magnetization and the nematic order parameter are

$$
\begin{aligned}
\langle g'|\sigma_j^x|g'\rangle &= \langle g'_-|\Pi^x\sigma_j^x|g'_-\rangle, \\
\langle g'|O_j|g'\rangle &= \langle g'_-|\Pi^x O_j|g'_-\rangle.
\end{aligned}
\tag{45}
$$

These site dependent expectation values can present a complex pattern, from which the behavior in the thermodynamic limit might not be obvious. Hence, following [16] we resort to their Discrete Fourier Transform (DFT)

$$
\mathcal{T}_K \equiv \frac{1}{N}\sum_{j=1}^{N}\langle g'|K_j|g'\rangle\, e^{\frac{2\pi i}{N}kj}, \quad k=1,\ldots,N,
\tag{46}
$$

that allows a quantitative analysis of their behavior in the thermodynamic limit.

In Fig. 5 we focus on the analysis of the magnetization, presenting the results obtained for its DFT $\mathcal{T}_{\sigma^x}$ as a function of the inverse chain length. We see that for $\delta_x > 0$ (upper panel) all sampled values go towards finite values in the thermodynamic limit, hence reproducing the typical behavior of the DFT of the staggered AFM order [16]. On the contrary, changing the sign of $\delta_x$, the DFT goes to zero for all $k$, hence signaling zeroing of the magnetization independently of the site taken into account. The effect of a negative defect ($\delta_x < 0$) for $\phi > 0$ is to strengthen the AFM interaction, so reinforcing the topological frustration, while a positive defect ($\delta_x > 0$) weakens the Ising term, so reducing the effect of the frustration and, as a consequence, allowing the existence of a macroscopic phase characterized by a local

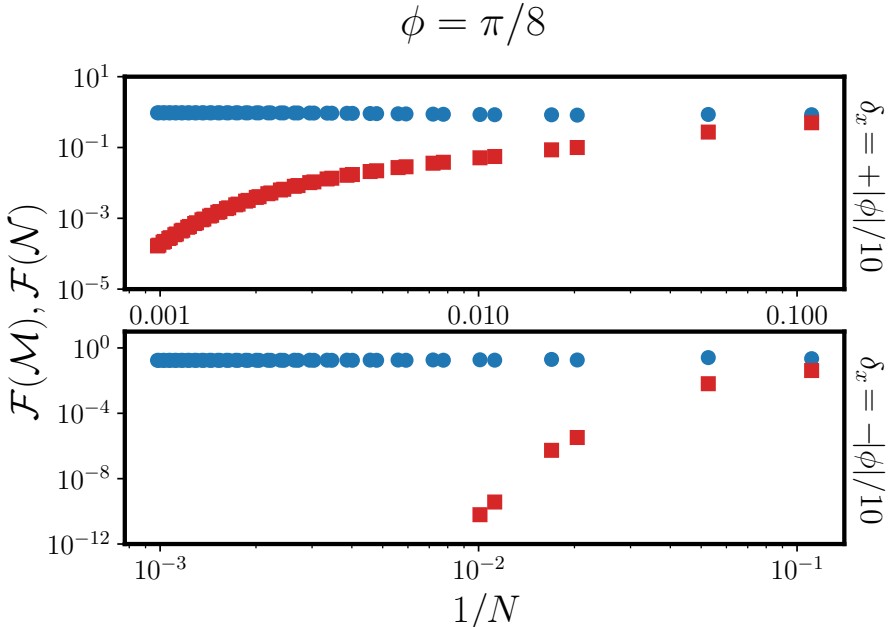

Figure 6: (Color online) Dependence of the absolute value of the ground state expectation values $\mathcal{F}(\mathcal{M})$ (blue circles) and $\mathcal{F}(\mathcal{N})$ (red squares), for the string operators defined in eq. (23), on the inverse chain length, for chain lengths up to $N = 1019$, at $\phi = \frac{\pi}{8}$. We observe that, while $\mathcal{F}(\mathcal{M})$ tends to a finite value, $\mathcal{F}(\mathcal{N})$ goes to zero. For both types of defects we have thus qualitatively a behavior as in Fig. 4. The exact asymptotic value for large $N$ depends on $\delta_x$: $\mathcal{F}(\mathcal{M}) \simeq 0.95$ and $\mathcal{F}(\mathcal{M}) \simeq 0.17$ in the upper and lower panel respectively.

magnetic order parameter. This implies that, while it is possible to remove the peculiar phase that we have found by the presence of a localized defect as the one we have considered, this fact depends on its sign and hence our results are, at least partially, resilient to the presence of a defect.

To further strengthen this result in Fig. 6 we have also analyzed the behavior of the ground state expectation values of the two string order operators, $\mathcal{F}(\mathcal{M}) \equiv \langle g' | \mathcal{M} | g' \rangle = \langle g'_- | \mathcal{M} | g'_- \rangle$ and $\mathcal{F}(\mathcal{N}) \equiv \langle g' | \mathcal{N} | g' \rangle = \langle g'_- | \mathcal{N} | g'_- \rangle$, as a function of $1/N$. In the figure we can appreciate that, regardless of the sign of the defect, $\mathcal{F}(\mathcal{M})$ remains finite in the thermodynamic limit. This result is in strong agreement with the fact that the phases discovered in the 2-Cluster-Ising model are partially resilient to the presence of a localized defect.

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
