# Peer review of "Topological Frustration can modify the nature of a Quantum Phase Transition"

_SciPost Physics, doi:SciPost Phys. 12, 075 (2022)_

## Round 3 · Referee Report · Anonymous (Referee 1) · 2021-7-14

Strengths

1) Interesting results 2) High scientific standards 3) Well written

Weaknesses

1) I personally could not estrapolate a general concept. I think that a hand waving argument from 'the more mathematical companion article' would improve the present paper. In other words, which are the models that are 'dangerous'? I understand that to answer this question some rigor could be lost, but I think that even a qualitative answer would make the article more fruitful in terms of stimulating further investigations.

Report

The Authors challenge some general principles behind the Ginzburg Landau theory. More specifically, they show that the order parameter can change from local to string-like for an appropriate choice of the boundary conditions. The model that they employ is solvable via a Wigner Jordan transformation , and a machinery similar to the one of the xy chain can be employed. The key ingredient is the adoption of 'frustrated boundary conditions'.
The results are interesting, well supported, and well explained.

Requested changes

I would appreciate the inclusion of a simple intuitive argument, if available.

  • validity: top
  • significance: good
  • originality: high
  • clarity: top
  • formatting: excellent
  • grammar: excellent

Author:  Fabio Franchini  on 2021-10-16  [id 1856]

(in reply to Report 1 on 2021-07-14)
Category:
remark
answer to question

We thank the referee for their positive appraisal of our work.
As we discuss in the new version of the manuscript, the general picture behind our results can be appreciated starting with the single excitation interpretation of FBC. Namely, the ground state of a quantum chain with FBC can be characterized by a single delocalized excitation, with lattice momentum p. In its motion, this excitation flips every spin and thus tends to destroy any order. Only if the ground state manifold contains several such states and only if two of them have momenta differing by $\pi$ in the thermodynamic limit, then the system can support a finite (although non-translational invariant) order, due to a constructive interference between the two spinon waves. Otherwise, any local order in the system vanishes for sufficiently large chain lengths. As it turns out, models with only nearest neighbor interaction, whose frustration is induced purely by the boundary conditions (such as the XYZ chain), tend to host ground states with momenta close to $\pm \pi/2$ and thus to support a finite order. However, when more general interactions are considered, they can create effective closed loops which increase the degree of frustration in the system (as it happens with the ANNNI model) and in these cases the ground states do not meet the requirement to have a finite order and thus can behave like the exactly solvable model we have considered.

---

## Round 3 · Referee Report · Anonymous (Referee 2) · 2021-8-16

Strengths

The paper is well written, rigorous, presents interesting results which can be highly relevant to researchers working on phase transitions.

Weaknesses

The authors derive all the results for a specific model, given by Eq. (1). However, the authors claim that the results are more general, and refer to reference [18] for the proof. I suggest the authors elaborate more on an intuitive / hand waiving argument regarding why these behaviors can be expected in models other than the one described by Eq. (1).

Report

The authors show that topological frustration in finite size spin systems, arising due to frustrated boundary conditions, can lead to a critical point separating two phases with zero local order parameters. This is in contrast to conventional phase transitions, where a critical point is separated by two phases with different local order parameters. However, here the phases on either side of the critical point are characterized by distinct topological order parameters.
This work is very interesting and revisits the problem of quantum phase transitions from a new perspective. It shows that since in practice all systems are finite sized, so frustrated boundary conditions may become highly relevant close to phase transitions.

Requested changes

I suggest the authors elaborate more on an intuitive / hand waiving argument regarding why these behaviors can be expected in models other than the one described by Eq. (1). For example, what are the general characteristics of the models in which these kind of behaviors can be expected?

  • validity: high
  • significance: high
  • originality: high
  • clarity: high
  • formatting: excellent
  • grammar: perfect

Author:  Fabio Franchini  on 2021-10-16  [id 1857]

(in reply to Report 2 on 2021-08-16)

We thank the referee for their positive appraisal of our work.
As we discuss in the new version of the manuscript, the general picture behind our results can be appreciated starting with the single excitation interpretation of FBC. Namely, the ground state of a quantum chain with FBC can be characterized by a single delocalized excitation, with lattice momentum p. In its motion, this excitation flips every spin and thus tends to destroy any order. Only if the ground state manifold contains several such states and only if two of them have momenta differing by $\pi$ in the thermodynamic limit, then the system can support a finite (although non-translational invariant) order, due to a constructive interference between the two spinon waves. Otherwise, any local order in the system vanishes for sufficiently large chain lengths. As it turns out, models with only nearest neighbor interaction, whose frustration is induced purely by the boundary conditions (such as the XYZ chain), tend to host ground states with momenta close to $\pm \pi/2$ and thus to support a finite order. However, when more general interactions are considered, they can create effective closed loops which increase the degree of frustration in the system (as it happens with the ANNNI model) and in these cases the ground states do not meet the requirement to have a finite order and thus can behave like the exactly solvable model we have considered.

---

## Round 4 · Referee Report · Anonymous (Referee 2) · 2021-10-25

Report

The authors have addressed the comments included in my first report and modified the manuscript accordingly. I recommend publication of this manuscript.

---

## Round 4 · Referee Report · Anonymous (Referee 1) · 2021-11-26

Report

All questions have been convincingly addressed by the authors and I hence recommend the publication of the paper.

---

## Round 4 · Author Response

We thank the editor and the referees for their work and positive assessment of our manuscript.
Both reports appreciated our work and its results and the only request, in both of them, was for some general arguments to better understand the generality of our findings.
We apologize it took us a long time to answer this inquiry and provide a new version of the manuscript, but this question motivated us for a more detailed study of the content of the "companion more mathematical paper". We have now updated substantially the analysis we performed in the latter work and this enabled us to better answer the referee's request. We have added a couple of paragraphs to the first version of the manuscript explaining the general (qualitative) picture.
We think that with these changes our work should be suitable for acceptance and publication.

---

## Round 4 · List of Changes

• The first paragraph in the first column of page 2 contains a new explanation of the qualitative picture underlying our results;
  • We added a new sentence at the end of the first Conclusion paragraph ending at the beginning of page 5;

---

## Editorial Decision

published